# OpenReview forum: "UniARM: Towards a Unified Autoregressive Reward Model for Multi-Objective Test-Time Alignment"
_ICLR.cc/2026/Conference — Submitted to ICLR 2026_

### Official Review · Reviewer_97Hm · 2025-10-24

**Soundness:** 2
**Presentation:** 3
**Contribution:** 2
**Rating:** 4
**Confidence:** 4

**Summary:**

This paper targets multi-objective alignment, and proposes Preference-Modulated & Shared Low-Rank Adaptation (MoSLoRA) for autoregressive reward modelling. Specifically, MoSLoRA extracts features shared across preferences and then adopts affine transformation to shared features. Building upon MoSLoRA, this paper proposes the UniARM, a autoregressive reward modelling framework for multi-objective test-time alignment. Experiments demonstrate the effectiveness of UniARM in safety and helpfulness alignment tasks, outperforming existing baselines.

**Strengths:**

- This paper addresses an interesting direction of multi-objective test-time alignment.
- This paper proposes a new MoSLoRA architecture.
- Experiments show the effectiveness of the proposed MoSLoRA and UniARM.
- There is a clear structure in related work review.
- The case study presentation is very intuitive.

**Weaknesses:**

- It seems that MoSLoRA relies on predefined semantic representation $o$, whereas PBLoRA does not. Therefore, it is unclear whether the performance improvement of MoSLoRA over PBLoRA stems from its architecture that integrates more information.
- It seems that this work is an incremental research study of previous PARM, and MoSLoRA is an improved version of PBLoRA. While many readers are familiar with standard LoRA, they may be less familiar with PARM and its proposed PBLoRA. I think that additional explanation or background from standard LoRA will enhance the readability of the paper.
- The evaluation metrics used in this paper are mainly HV and MIP. However, WinRate is a more widely used metric in the LLM preference alignment studies, but it is not involved into this paper to assess the generation quality of the model.
- It seems that the experiments are primarily conducted on Alpaca-7B, which is a relatively outdated model. Using more recent models, such as Qwen3 series, will be better convincing.
- There is no anonymous code link to present the reproducibility of this work.

**Questions:**

- Have you considered comparing UniARM with vanilla RLHF methods, such as PPO with a pretrained reward model, under test-time alignment scenario?

---

> ### Author Response · Authors · 2025-11-19
>
> We sincerely thank Reviewer 97Hm for the detailed and insightful feedback.  Below, we provide detailed responses to the questions and concerns raised.
>
> > **Weaknesses 1:** It seems that MoSLoRA relies on predefined semantic representation , whereas PBLoRA does not. Therefore, it is unclear whether the performance improvement of MoSLoRA over PBLoRA stems from its architecture that integrates more information.
> >
>
> We thank the reviewer for this insightful observation. We would like to clarify that the “semantic representations” relied upon by MoSLoRA are not external information but are entirely derived from the embedding layer of the ARM model. Therefore, while MoSLoRA and PBLoRA [1] are placed in the same experimental setting, **MoSLoRA is explicitly designed to leverage the intrinsic information of the ARM, whereas PBLoRA does so implicitly.**
>
> The performance improvement of MoSLoRA stems from our innovative architectural design, which enables ARM to fully leverage its intrinsic information to more effectively distinguish between different preference objectives. In contrast, PBLoRA lacks a mechanism to fully utilize this intrinsic information. We consider this architectural innovation, which allows ARM to fully exploit its intrinsic information, one of the key contributions of our work.
>
> Obtaining these  semantic representations is straightforward and can be accomplished in just two lines of code.
>
> ```python
> object_input_ids = tokenizer(object_text_description)
> obj_embed = torch.mean(model.base_model.get_input_embeddings()(object_input_ids), dim=0)
> ```
>
> > **Weaknesses 2:** It seems that this work is an incremental research study of previous PARM, and MoSLoRA is an improved version of PBLoRA. While many readers are familiar with standard LoRA, they may be less familiar with PARM and its proposed PBLoRA. I think that additional explanation or background from standard LoRA will enhance the readability of the paper.
> >
>
> We would like to clarify that MoSLoRA is not merely an incremental improvement over PBLoRA. **We would kindly like to refer you to Concern 1 of our global response, where we compare MoSLoRA and PBLoRA and provide an intuitive explanation**.
>
> We sincerely thank the reviewer for the kind reminder. **We have reviewed related works such as PBLoRA and MTLoRA in Appendix B (Additional Related Work)** and discussed the distinctions between these methods and our approach. In the revised manuscript, we will also include a brief introduction to standard LoRA to improve readability for readers who are less familiar with PARM and PBLoRA.
>
> > **Weaknesses 3:** The evaluation metrics used in this paper are mainly HV and MIP. However, WinRate is a more widely used metric in the LLM preference alignment studies, but it is not involved into this paper to assess the generation quality of the model.
> >
>
> We fully agree that WinRate is the gold standard for evaluating the quality of a single solution in LLM preference alignment studies, especially in single-objective scenarios. **However, the focus of this work is on a multi-objective alignment problem. In such problems, our goal is not to generate a single “best” response, but to produce a Pareto solution set that represents a set of optimal trade-offs among different conflicting objectives (e.g., helpfulness vs. harmlessness). The design philosophy of WinRate is fundamentally inconsistent with this goal.**
>
> WinRate is a pairwise comparison metric intended to select a “winner” between two candidates. Therefore, WinRate is not suitable for assessing two key quality dimensions of a solution set:
>
> - **Convergence**: how close the solution set is to the true Pareto-optimal frontier.
> - **Diversity**: whether the solution set sufficiently and uniformly covers all possible trade-off directions.
>
> **In contrast, the Hypervolume (HV) and Mean Inner Product (MIP) metrics we adopt are widely recognized in multi-objective  studies [1-7] as standard measures specifically designed to evaluate the quality of Pareto solution sets.**
>
> > **Weaknesses 4:** It seems that the experiments are primarily conducted on Alpaca-7B, which is a relatively outdated model. Using more recent models, such as Qwen3 series, will be better convincing.
> >
>
> We sincerely thank the reviewer for the kind reminder. We have supplemented experiments on other latest model families (Qwen-3 and Tulu-3) across different tasks and would **kindly like to refer you to Concern 2 of our global response for further details**.
>
> > **Weaknesses 5:** There is no anonymous code link to present the reproducibility of this work.
> >
>
> We would like to clarify that the code for this work is included in the **supplementary material**. This is also stated in the “**Reproducibility Statement**” of the submitted manuscript. In the future, we plan to provide a public code repository link.

---

> > ### Author Response · Authors · 2025-11-19
> >
> > **Questions**
> > >Have you considered comparing UniARM with vanilla RLHF methods, such as PPO with a pretrained reward model, under test-time alignment scenario?
> > >
> >
> > **We would like to clarify that a direct comparison between UniARM and standard RLHF methods is not entirely appropriate.** UniARM is designed for multi-objective test-time alignment, allowing the generation of a Pareto solution set by adjusting user preference, whereas standard RLHF methods typically produce only a single solution. Therefore, the two approaches differ fundamentally in their design goals and applicable scenarios. The recently proposed MORLHF approach RS [8], a multi-objective variant of standard RLHF, can be used to train multi-objective policy models. **We have already compared UniARM with MORLHF approach RS in Table 1 of the submitted manuscript.**
> >
> > [1] Lin B, Jiang W, Xu Y, et al. PARM: Multi-Objective Test-Time Alignment via Preference-Aware Autoregressive Reward Model[J]. ICML, 2025.
> >
> > [2] Zhang X, Zhao L, Yu Y, et al. LibMOON: A gradient-based multiobjective optimization library in PyTorch[J]. Advances in Neural Information Processing Systems, 2024, 37: 2026-2044.
> >
> > [3] Mukherjee S, Lalitha A, Sengupta S, et al. Multi-objective alignment of large language models through hypervolume maximization[J]. arXiv preprint arXiv:2412.05469, 2024.
> >
> > [4] Yao S, Liu F, Lin X, et al. Multi-objective evolution of heuristic using large language model[C]//Proceedings of the AAAI Conference on Artificial Intelligence. 2025, 39(25): 27144-27152.
> >
> > [5] Ren Y, Xiao T, Shavlovsky M, et al. COS-DPO: Conditioned One-Shot Multi-Objective Fine-Tuning Framework[C]//The 41st Conference on Uncertainty in Artificial Intelligence.
> >
> > [6] Hayes C F, Rădulescu R, Bargiacchi E, et al. A practical guide to multi-objective reinforcement learning and planning[J]. Autonomous Agents and Multi-Agent Systems, 2022, 36(1).
> >
> > [7] Huang Y, Wu S, Zhang W, et al. Autonomous multi-objective optimization using large language model[J]. IEEE Transactions on Evolutionary Computation, 2025.
> >
> > [8] Rame A, Couairon G, Dancette C, et al. Rewarded soups: towards pareto-optimal alignment by interpolating weights fine-tuned on diverse rewards[J]. NIPS, 2023.

---

> ### Author Response · Authors · 2025-11-27
>
> We sincerely thank the reviewer for your time and effort in reviewing our paper! We have addressed the specific points you raised regarding  aspects of our methodological improvements, experimental evaluation, reproducibility, etc., along with questions about our work.
>
> Please let us know whether we have addressed your concerns. We are more than happy to provide additional clarifications if you have further questions. Thank you!

---

> ### Comment · Reviewer_97Hm · 2025-11-27
> **Response to authors**
>
> Thanks for the authors' detailed response.
>
> I understand that the goals of multi-objective alignment and standard preference alignment are different. However, in fact, the task of multi-objective alignment can be viewed as an extension of standard preference alignment. The authors stated that "a direct comparison between UniARM and standard RLHF methods **is not entirely appropriate**." I am not sure I agree with that. As multi-objective alignment is essentially built upon the standard alignment setting, my view is that certain comparisons would help clearly demonstrate the advantages of multi-objective alignment over the standard alignment problem. Such comparisons could also clarify the motivation for defining and addressing this new task, and explain under what scenarios multi-objective alignment is needed. As we know, preference alignment is an important stage in the LLM training pipeline, and this naturally drives substantial research effort to improve its algorithms and enhance model performance.
>
> At present, I feel that the authors introduce a new problem from an existing one and propose a method to address it, but the connection between the two is not sufficiently established. The authors in this paper attempts to persuade readers to accept the new problem, but authors seems to avoid necessary comparisons with the existing problem and present that comparisons "are not entirely appropriate," which further creates a sense of disconnect.
>
> All in all, I choose to maintain my current rating of 4. Considering the contributions on the proposed methods, I leave the final decision to the AC.
>
> Good luck to the authors.

---

> ### Author Response · Authors · 2025-11-27
>
> We apologize for any confusion caused by our previous response. We would like to clarify that **we have not avoided performing the necessary comparisons with standard RLHF**. It should be noted that standard RLHF alignment is a **single-objective alignment**, which can essentially be regarded as a **subset of the multi-objective alignment problem**. When the **harmlessness weight is 0 and the helpfulness weight is 1**, or when the **harmlessness weight is 1 and the helpfulness weight is 0**, all methods can be regarded as **single-objective alignment**, **in which case RS is equivalent to standard RLHF**. This comparison is already presented in Figure 3(a) of our submitted manuscript. To make the comparison clearer, we provide the corresponding scores in the table below:
>
> | **Method** | **Harmlessness Weight** | **Helpfulness Weight** | **Harmlessness Score** | **Helpfulness Score** | Avg Score  |
> | --- | --- | --- | --- | --- | --- |
> | RS (Standard RLHF) | 0 | 1 | -12.51 | 5.04 | -3.74 |
> | Our Method | 0 | 1 | -10.14 | 4.80 | -2.67 |
> | RS (Standard RLHF) | 1 | 0 | 5.37 | -2.33 | 1.52 |
> | Our Method | 1 | 0 | 9.32 | 0.22 | 4.77 |
>
> When aligning only helpfulness (harmlessness weight = 0, helpfulness weight = 1), our method achieves comparable performance to standard RLHF in terms of helpfulness score, while surpassing it in harmlessness score, resulting in an average score improvement of approximately **40%**.
>
> When aligning only harmlessness (harmlessness weight = 1, helpfulness weight = 0), our method outperforms standard RLHF in both helpfulness and harmlessness scores, achieving an average score improvement of approximately **213%**.
>
> It should be noted that, since standard RLHF is a single-objective alignment, we can only compare it at the **extreme points** (i.e., when either the harmlessness weight or the helpfulness weight is 0 or 1).
>
> Thank you again for your time and effort in reviewing our paper! Please let us know if the above explanations do not address your concerns. We are happy to answer any further questions.

---

> > ### Comment · Reviewer_97Hm · 2025-11-28
> > **Response to authors**
> >
> > Thanks for the authors' detailed explanation.
> >
> > In general, aligning LLM responses often requires pursuing multiple objectives, such as the HHH (helpful, harmless, honest) principle. There also exist reward models explicitly trained for multi-objective reward modeling, such as ArmoRM-Llama3-8B-v0.1 [1], which has been widely adopted in many recent studies. I still believe that important comparisons and discussions with RL training using such reward models are missing. If these results were included, I would be inclined to raise my score; however, I am unable to update the score due to the accident.
> >
> > Good luck to authors.
> >
> > [1] Wang H, Xiong W, Xie T, et al. Interpretable Preferences via Multi-Objective Reward Modeling and Mixture-of-Experts[C]//Findings of the Association for Computational Linguistics: EMNLP 2024. 2024: 10582-10592.

---

> > > ### Author Response · Authors · 2025-11-30
> > >
> > > We sincerely thank the reviewer for the kind reminder. We have supplemented experiments on ArmoRM-Llama3-8B-v0.1 and conducted comparisons and discussions; and would kindly like to refer you to Concern 3 of our global response for further details.
> > >
> > > ---
> > >
> > > Thank you again for your time and effort in reviewing our paper! Please let us know if the above explanations do not address your concerns. We are happy to answer any further questions.

---

### Official Review · Reviewer_rgLD · 2025-10-28

**Soundness:** 3
**Presentation:** 3
**Contribution:** 3
**Rating:** 6
**Confidence:** 4

**Summary:**

This paper proposes UniARM, a unified framework for multi-objective alignment of LLMs that balances multiple human preference goals at test time. It achieves this by introducing MoSLoRA, which learns shared features across preferences and modulates them through preference-conditioned transformations, reducing feature entanglement and enabling fine-grained trade-offs. The approach improves alignment quality and efficiency without slowing inference and without retraining the base LLMs over and over again.

**Strengths:**

1. This paper deals with a very relevant area of using test-time alignment method to achieve multi-objective goals and achieve good experimental result.

2. Unlike the prior art PARM, this method does not require linearly combining different core tensors based on the preference vector during test time, which is more reasonable.

3. The experiments are extensive, including helpfulness and harmlessness evaluation, as well as weak-to-strong extension. It is especially good to see the weak-to-strong experiments as it is very expensive to retrain larger LLMs for multi-objective goals using training-based method, and the proposed method seems like an efficient alternative.

4. The paper is well-written and clearly explain the difference between the prior work.

**Weaknesses:**

1. While the experiments setting follows prior work, the LLM used here seems not very up to date. It would be nice if the author can evaluate on more recent LLMs. For example, tulu-3 instead of tulu-2.

Other minor issues
1. Typo in equation (10)
2. In Figure 2, the results of RS and MOD are set to zero. Although it is understandable that they are very expensive to run, I am not sure if it is a good idea to set them to be zero.

**Questions:**

If the model has been trained for, for example, objective A and B, and then there is a different objective C, what is the quickest way to adapt to the new objective? It seems that in GenARM, the reward model of objective C can be immediately used. I wonder what we should do for your proposed UniARM.

---

> ### Author Response · Authors · 2025-11-19
>
> We sincerely thank Reviewer rgLD for the detailed and insightful feedback.  Below, we provide point-by-point responses to the reviewer’s questions.
>
> > **Weaknesses 1:** While the experiments setting follows prior work, the LLM used here seems not very up to date. It would be nice if the author can evaluate on more recent LLMs. For example, tulu-3 instead of tulu-2.
> >
>
> We sincerely thank the reviewer for the kind reminder. We have supplemented experiments on other latest model families (Qwen-3 and Tulu-3) across different tasks and **would kindly like to refer you to Concern 2 of our global response for further details**.
>
> **Other minor issues**
>
> > Typo in equation (10)
> >
>
> We sincerely thank the reviewer for the careful review. We have already revised this in the updated manuscript.
>
> > In Figure 2, the results of RS and MOD are set to zero. Although it is understandable that they are very expensive to run, I am not sure if it is a good idea to set them to be zero.
> >
>
> Although RS and MOD were temporarily set to zero in the weak-to-strong experiments due to limited training resources, the quantitative results in the safety alignment experiments already demonstrate the superiority of our approach over RS[1] and MOD[2]. Moreover, in the weak-to-strong experiments, we also show that our approach achieves significant improvements compared to the latest multi-objective test-time alignment approaches(GenARM[3], PARM[4]) . We believe these results sufficiently demonstrate the effectiveness of our approach.  Additionally,  we are seeking funding and computational support to address this issue as soon as possible.
>
> **Questions**
> > If the model has been trained for, for example, objective A and B, and then there is a different objective C, what is the quickest way to adapt to the new objective? It seems that in GenARM, the reward model of objective C can be immediately used. I wonder what we should do for your proposed UniARM.
> >
>
> We would like to clarify that when a new objective C emerges, GenARM requires training a separate ARM for objective C. We also need to retrain the ARM for objective C. The key difference is that during inference, our method requires only a single ARM, whereas GenARM requires a separated ARM for each objective.
>
> [1] Rame A, Couairon G, Dancette C, et al. Rewarded soups: towards pareto-optimal alignment by interpolating weights fine-tuned on diverse rewards[J]. NIPS, 2023.
>
> [2] Shi R, Chen Y, Hu Y, et al. Decoding-time language model alignment with multiple objectives[J]. NIPS, 2024.
>
> [3] Xu Y, Sehwag U M, Koppel A, et al. GenARM: Reward Guided Generation with Autoregressive Reward Model for Test-Time Alignment[C]. ICLR, 2025.
>
> [4] Lin B, Jiang W, Xu Y, et al. PARM: Multi-Objective Test-Time Alignment via Preference-Aware Autoregressive Reward Model[J]. ICML, 2025.

---

> > ### Comment · Reviewer_rgLD · 2025-11-22
> >
> > I appreciate the efforts on running experiments on more recent models, and my concerns are addressed.

---

> > > ### Author Response · Authors · 2025-11-26
> > >
> > > Thank you for your response! We are glad that your concerns have been addressed.

---

### Official Review · Reviewer_9Eti · 2025-10-30

**Soundness:** 2
**Presentation:** 3
**Contribution:** 2
**Rating:** 4
**Confidence:** 3

**Summary:**

This paper introduces UniARM, a framework for multi-objective test-time alignment. The proposed method employs a parameter-efficient architecture called MoSLoRA, which consists of a preference-agnostic module and a preference-modulation module. Experimental results demonstrate the effectiveness of UniARM compared to existing test-time alignment approaches.

**Strengths:**

- The related work section is comprehensive.

- The method eliminates the need to train multiple separate ARMs or preference-aware modules. Instead, it requires training only one preference-agnostic module and one preference-modulation module for alignment multiple objectives.

- The experimental results show superior performance compared to previous test-time alignment methods.

**Weaknesses:**

Overall, I find the methodological design of the paper reasonable, though the experimental section requires further strengthening. If the following concerns can be adequately addressed, I would consider raising my score.

- I'm unfamiliar with test-time multi-objective alignment methods based on ARM. Therefore, I'm puzzled about the motivation behind this approach. Since we can fine-tune the ARM, why do we just fine-tune the LLM itself?

- The backbone model used is relatively outdated. Employing a more recent backbone model (e.g., LLaMA-3, Qwen-3) and reward model (e.g., ARMO-RM, Skywork-LLaMA-V2) would strengthen the validity of the results.

- Since the ARM also involves fine-tuning the base model, it is essential to compare UniARM with other training-based multi-objective methods, such as MODPO and other state-of-the-art baselines.

- Incorporating LLM-as-a-judge evaluation on benchmarks like AlpacaEval (for helpfulness) would provide a more realistic and convincing assessment.

- Table 2: It would be beneficial to include results without any ARM, using only Alpaca-65B. Additionally, the use of an older model raises concerns about whether similar performance can be achieved with newer backbone models.

- An analysis of the sensitivity to hyperparameters such as λ and β is missing. Also, the impact of varying the text descriptions of objectives on the results should be investigated.

- Details regarding training and inference computational costs are not provided.

Typos:

- Line 32: The meaning of "PB" should be explicitly explained to enhance clarity.

- Eq. (3):The notation "i" is not defined, which may cause confusion.

- Eq. (10): A closing bracket ")" is missing.

**Questions:**

See weakness.

---

> ### Author Response · Authors · 2025-11-19
>
> We sincerely thank Reviewer 9Eti for the detailed feedback. Below, we provide detailed responses to these comments.
>
> > **Weaknesses 1:** I'm unfamiliar with test-time multi-objective alignment methods based on ARM. Therefore, I'm puzzled about the motivation behind this approach. Since we can fine-tune the ARM, why do we just fine-tune the LLM itself?
> >
>
> Multi-objective test-time alignment enables frozen **large** language models (LLMs) to adapt to multidimensional user preferences during inference by guiding their generation process with **small** autoregressive reward models (ARMs).
>
> In a multi-objective setting, user preferences vary across individuals and evolve over time, meaning that **training a separate large policy model (e.g., Alpaca-65B) for every possible preference combination would significantly increase computational costs and  require maintaining a large number of independent models**. Therefore, it is practically infeasible for most research teams. To address this, UniARM leverages a smaller ARM (e.g., Alpaca-7B) to guide the generation process of the frozen base LLM (e.g., Alpaca-65B) , enabling it to handle all objective combinations without fine-tuning the LLM itself. Consequently, multi-objective test-time alignment provides a practical and scalable solution.
>
> > **Weaknesses 2:** The backbone model used is relatively outdated. Employing a more recent backbone model (e.g., LLaMA-3, Qwen-3) and reward model (e.g., ARMO-RM, Skywork-LLaMA-V2) would strengthen the validity of the results.
> >
>
> W2.1 **Regarding more recent backbone models**
>
> We sincerely thank the reviewer for the kind reminder. We have supplemented experiments on other latest model families (Qwen-3 and Tulu-3) across different tasks and **would kindly like to refer you to Concern 2 of our global response for further details**.
>
> **W2.2 Regarding more recent  reward model**
>
> **We would like to clarify that the reward models we adopt are widely used in multi-objective alignment studies** [1-9]. The latest reward models, such as ARMO-RM [10] and Skywork-LLaMA-V2 [11], do not improve the reliability or validity of our experiments for the following reasons:
>
> 1. Although ARMO-RM is a multi-objective reward model, its public evaluations report only a single scalar score obtained by compressing the multi-dimensional rewards, without assessing performance on individual objectives. Therefore, its effectiveness on each specific objective remains unclear.
> 2. Skywork-LLaMA-V2 is a general-purpose reward model that does not explicitly model separate preference objectives and cannot provide reward scores for individual objectives, making it unsuitable for our multi-objective preference alignment evaluation.
>
> Therefore, these two reward models are not suitable for our experimental setting. We will make the corresponding revisions in the manuscript.
>
> > **Weaknesses 3:** Since the ARM also involves fine-tuning the base model, it is essential to compare UniARM with other training-based multi-objective methods, such as MODPO and other state-of-the-art baselines.
> >
>
> **We would like to clarify that the design philosophy of multi-objective test-time alignment is to guide the generation of frozen base models by training the ARM, rather than fine-tuning the base model itself.** This approach allows flexible adaptation to different preference combinations while preserving the capabilities of the base model, thereby reducing the training cost of multi-objective alignment. Therefore, UniARM does not involve any fine-tuning of the base model.
>
> We agree that comparisons with MODPO[3] and other PPO/DPO-based multi-objective alignment methods are necessary. These methods typically require fine-tuning the policy model $\pi$ to accommodate different preference weights. Repeated fine-tuning of such policy models in a multi-objective setting poses a significant computational challenge.
>
> For example, in the weak-to-strong experiment of the Safety Alignment task (ARM 7B, base LLM 65B), MODPO would need to train an independent 65B policy model for each preference weight combination (e.g., $\alpha$ ranging from 0 to 1 in increments of 0.1), resulting in a total of 11 separate 65B policy models. Due to limited computational resources, we are currently unable to perform this extremely costly comparison.
>
> In contrast, UniARM only requires training a single 7B reward model to handle all preference weight combination and guide the generation of the frozen 65B base model. This significantly improves computational efficiency and makes multi-objective alignment feasible within practical resource budgets for most researchers.

---

> ### Author Response · Authors · 2025-11-19
>
> > **Weaknesses 4:** Incorporating LLM-as-a-judge evaluation on benchmarks like AlpacaEval (for helpfulness) would provide a more realistic and convincing assessment.
> >
>
> We fully agree that the LLM-as-a-judge approach for computing WinRate is the gold standard for evaluating the quality of a single solution in LLM preference alignment studies. **However, the focus of this work is on a multi-objective alignment problem. In such problems, our goal is not to generate a single “best” response, but to produce a Pareto solution set that represents a set of optimal trade-offs among different conflicting objectives (e.g., helpfulness vs. harmlessness). The design philosophy of WinRate is fundamentally inconsistent with this goal.**
>
> WinRate is a pairwise comparison metric intended to select a “winner” between two candidates.  Therefore, WinRate is not suitable for assessing two key quality dimensions of a solution set:
>
> - **Convergence**: how close the solution set is to the true Pareto-optimal frontier.
> - **Diversity**: whether the solution set sufficiently and uniformly covers all possible trade-off directions.
>
> In contrast, the Hypervolume (HV) and Mean Inner Product (MIP) metrics **we adopt are widely recognized in multi-objective studies [2,12-17]** as standard measures specifically designed to evaluate the quality of Pareto solution sets.
>
> > **Weaknesses 5:**  Table 2: It would be beneficial to include results without any ARM, using only Alpaca-65B. Additionally, the use of an older model raises concerns about whether similar performance can be achieved with newer backbone models.
> >
>
> We agree with the reviewer that comparing against the base model (Alpaca-65B without ARM) is crucial. **This comparison is indeed included in our manuscript in Figure 3(b),** where the "base model" represents Alpaca-65B without ARM.  For example, when the user preference weights are Harmlessness 0.9 and Helpfulness 0.1, the base model guided by UniARM outperforms the unguided base model. Similarly, many other weight combinations also show superior performance compared to the base model without ARM guidance.
>
>  Regarding the request to add this baseline to Table 2, we would like to respectfully clarify that the metrics in Table 2, Hypervolume (HV) and Mean Inner Product (MIP), are specifically designed to evaluate the quality and diversity of a Pareto set (a set of solutions) generated by a multi-objective optimization method. The base Alpaca-65B model, when run without ARM, produces only a single solution (one response) for a given prompt, not a set of solutions. Therefore, the HV and MIP metrics are mathematically inapplicable to this single-point baseline.
>
> Regarding the performance of newer backbone models. We have supplemented experiments on other latest model families (Qwen-3 and Tulu-3) across different tasks and **would kindly like to refer you to Concern 2 of our global response for further details**.

---

> ### Author Response · Authors · 2025-11-19
>
> > **Weaknesses 6:**  An analysis of the sensitivity to hyperparameters such as λ and β is missing. Also, the impact of varying the text descriptions of objectives on the results should be investigated.
> >
>
> We sincerely thank the reviewer for the kind reminder. We would like to clarify that an **ablation study** on λ has already been included in the submitted manuscript, where we analyze its effect.
>
> We have now included a hyperparameter sensitivity analysis for β on the Safety Alignment task. As shown in Table 1, setting β to 0.01 yields the best performance.
>
> Table 1: Hyperparameter sensitivity analysis of β on the safety alignment task
>
> | β  | HV | MIP |
> | --- | --- | --- |
> | 0.1 | 81.67 | -0.74 |
> | **0.01** | **129.16** | **3.58** |
> | 0.001 | 83.69 | 1.94 |
>
> Meanwhile, to ensure a fair comparison across different methods and to avoid misleading improvements that arise from tuning the hyperparameter $\beta$ rather than from the methods themselves, we keep $\beta$ identical for our method and the baseline methods GenARM and PARM.
>
> We conducted an ablation study on the safety alignment task to evaluate the impact of different textual descriptions of objectives.  The text descriptions and corresponding experimental results are as follows:
>
> Table 2: Different textual descriptions of objectives
>
> | **Text Description** | **Harmless** | **Helpful** |
> | --- | --- | --- |
> | **Description 1** | The response should avoid content that is offensive, discriminatory, or harmful. | The response should provide useful resources and suggestions to the user. |
> | **Description 2** | The response is required to refrain from producing content that is offensive, discriminatory, or otherwise harmful. | The response should aim to assist the user by providing informative, actionable, and relevant suggestions. |
> | **Description 3** | Make sure the response is safe and free from anything offensive or harmful. | Provide the user with helpful tips, advice, or resources they can use. |
>
> Table 3: Impact of objective descriptions on UniARM performance in safety alignment
>
> |  | HV | MIP |
> | --- | --- | --- |
> | Text description1 | 129.16 | 3.58 |
> | Text description2 | 128.72 | 3.72 |
> | Text description3 | 129.52 | 3.62 |
>
> As shown in Table 3,  varying the textual descriptions has only a minimal effect on performance. As long as the descriptions sufficiently distinguish the semantics of different objectives, the resulting performance differences are almost negligible. Moreover, since our goal is not to meticulously engineer the objective descriptions.
>
> > **Weaknesses 7:** Details regarding training and inference computational costs are not provided.
> >
>
> We sincerely thank the reviewer for the kind reminder.  We would like to clarify that **Table 3 in the originally submitted manuscript provides** the number of trainable parameters for each approach, as well as the inference time required for each approach to guide the generation of the frozen base model.
>
> Additionally, we supplement here the training time required for fine-tuning the ARMs. Using a single NVIDIA A800 GPU, with batch size 32 and a total of 10k training samples, fine-tuning ARMs ( GenARM, PARM, and UniARM ) on the 7B model takes approximately 0.5 hours per epoch.
>
> **typos**
>
> > Line 32: The meaning of "PB" should be explicitly explained to enhance clarity.
> >
>
> We sincerely thank the reviewer for carefully reading our paper. We have checked Line 32 and confirm that the abbreviation “PB” does not appear there. We noticed that it first appears in Line 53, and we have added the corresponding explanation in the updated manuscript.  We appreciate the reviewer’s careful reading and will continue to ensure that all abbreviations are clearly defined throughout the manuscript.
>
> > Eq. (3):The notation "i" is not defined, which may cause confusion.
> >
>
> > Eq. (10): A closing bracket ")" is missing.
> >
>
> We have already corrected the above typos in the updated manuscript.

---

> > ### Author Response · Authors · 2025-11-19
> >
> > [1] Xu Y, Sehwag U M, Koppel A, et al. GenARM: Reward Guided Generation with Autoregressive Reward Model for Test-Time Alignment[C]. ICLR, 2025.
> >
> > [2] Lin B, Jiang W, Xu Y, et al. PARM: Multi-Objective Test-Time Alignment via Preference-Aware Autoregressive Reward Model[J]. ICML, 2025.
> >
> > [3] Zhou Z, Liu J, Shao J, et al. Beyond one-preference-fits-all alignment: Multi-objective direct preference optimization[C]. ACL, 2024.
> >
> > [4] Yang R, Pan X, Luo F, et al. Rewards-in-context: multi-objective alignment of foundation models with dynamic preference adjustment[C]//Proceedings of the 41st International Conference on Machine Learning. 2024: 56276-56297.
> >
> > [5] Li M, Zhang Y, Wang W, et al. Self-improvement towards pareto optimality: Mitigating preference conflicts in multi-objective alignment[J]. arXiv preprint arXiv:2502.14354, 2025.
> >
> > [6] Li Z, Du G, Guo W, et al. Multi-objective Large Language Model Alignment with Hierarchical Experts[J]. arXiv preprint arXiv:2505.20925, 2025.
> >
> > [7] Agnihotri A, Jain R, Ramachandran D, et al. Multi-Objective Preference Optimization: Improving Human Alignment of Generative Models[J]. arXiv preprint arXiv:2505.10892, 2025.
> >
> > [8] Gu H, Wang H, Mei Y, et al. ParetoHqD: Fast Offline Multiobjective Alignment of Large Language Models using Pareto High-quality Data[J]. arXiv preprint arXiv:2504.16628, 2025.
> >
> > [9]Son S, Bankes W, Yoon S, et al. Robust Multi-Objective Controlled Decoding of Large Language Models[C]//2nd Workshop on Models of Human Feedback for AI Alignment.
> >
> > [10]Wang H, Xiong W, Xie T, et al. Interpretable Preferences via Multi-Objective Reward Modeling and Mixture-of-Experts[C]//Findings of the Association for Computational Linguistics: EMNLP 2024. 2024: 10582-10592.
> >
> > [11] Liu C Y, Zeng L, Xiao Y, et al. Skywork-Reward-V2: Scaling Preference Data Curation via Human-AI Synergy[J]. arXiv preprint arXiv:2507.01352, 2025.
> >
> > [12] Zhang X, Zhao L, Yu Y, et al. LibMOON: A gradient-based multiobjective optimization library in PyTorch[J]. Advances in Neural Information Processing Systems, 2024, 37: 2026-2044.
> >
> > [13] Mukherjee S, Lalitha A, Sengupta S, et al. Multi-objective alignment of large language models through hypervolume maximization[J]. arXiv preprint arXiv:2412.05469, 2024.
> >
> > [14] Yao S, Liu F, Lin X, et al. Multi-objective evolution of heuristic using large language model[C]//Proceedings of the AAAI Conference on Artificial Intelligence. 2025, 39(25): 27144-27152.
> >
> > [15] Ren Y, Xiao T, Shavlovsky M, et al. COS-DPO: Conditioned One-Shot Multi-Objective Fine-Tuning Framework[C]//The 41st Conference on Uncertainty in Artificial Intelligence.
> >
> > [16] Hayes C F, Rădulescu R, Bargiacchi E, et al. A practical guide to multi-objective reinforcement learning and planning[J]. Autonomous Agents and Multi-Agent Systems, 2022, 36(1).
> >
> > [17] Huang Y, Wu S, Zhang W, et al. Autonomous multi-objective optimization using large language model[J]. IEEE Transactions on Evolutionary Computation, 2025.

---

> ### Author Response · Authors · 2025-11-27
>
> We sincerely thank the reviewer for your time and effort in reviewing our paper! We have addressed the specific points you raised regarding aspects of our motivation, experimental evaluation, etc.
>
> Please let us know whether we have fully addressed your concerns. We are more than happy to provide additional clarifications if you have further questions. Thank you!

---

> ### Comment · Reviewer_9Eti · 2025-11-28
>
> Sorry for my late reply.  I appreciate the authors' detailed responses and have carefully reviewed your rebuttal. The authors have addresses my other concerns. Here are some points I would like to further discuss with authors:
>
> > W2 Regarding more recent reward model
>
> Armo-RM [1] is actually widely used in existing researchs with ~250 citations up to date. I believe such comparsions would strength the paper.
>
> I would like to update my score once the authors attach the above experiments.
>
> References
>
> [1] Interpretable Preferences via Multi-Objective Reward Modeling and Mixture-of-Experts

---

> > ### Author Response · Authors · 2025-11-30
> >
> > We sincerely thank the reviewer for the kind reminder. We have supplemented experiments on ArmoRM-Llama3-8B-v0.1 and conducted comparisons and discussions; and would kindly like to refer you to Concern 3 of our global response for further details.
> >
> > ---
> >
> > Thank you again for your time and effort in reviewing our paper! Please let us know if the above explanations do not address your concerns. We are happy to answer any further questions.

---

### Official Review · Reviewer_SHwo · 2025-11-01

**Soundness:** 3
**Presentation:** 4
**Contribution:** 3
**Rating:** 8
**Confidence:** 4

**Summary:**

This paper tackles the problem of multi-objective test-time alignment of LLMs. The authors propose UniARM based on the better parameterization method named MoSLoRA. The method is parameter-efficient because it uses the semantic embeddings of the model itself to encode the multiple preferences instead of additional weights. At the same time, the empirical results also demonstrate the superiority of the method in comparison to previous state-of-the-art methods. Although there is some weakness, I believe they can be improved with minor revisions after the rebuttal. Therefore, I recommend "accept".

**Strengths:**

1. The paper is well-written and easy to understand.
2. The proposed method performs better while being more parameter-efficient than baselines. I believe this gain outweighs the slightly weak originality of the proposed method (weak originality as it consists in a minor change to the low-rank adapter used in PARM plus a regularizer).
3. I think the experiments are very well designed. Enough meaningful baselines are included. The ablation experiments provide clear clues as to how each component of the method affects the performance. The choice of the quantitative metric of HV and MIP makes the performance gap between the proposed method and baselines more convincing.
4. The method itself is simple to reimplement, which implies a strong potential to be an impactful contribution to the community.

**Weaknesses:**

1. Although the proposed method is empirically effective, there is a lack of intuitive or theoretical explanation of where the effectiveness (i.e., the better Pareto front) comes from. Can the authors explicitly provide such explanations? For example, why can a different parameterization of the token-level reward models alone (according to the ablation experiment when $\lambda=0$) lead to a better Pareto front?
2. (Partially related to the first weakness) The generality of the method is unknown. Is the better Pareto front only limited to the Alpaca family? Can the authors provide results on other families (Llama, Qwen, etc.) to make the generality of the method more convincing?
3. I find the term "Pareto-optimal" too strong and vague to describe an ARM as there is still room for improvement. I suggest a softer term "more/less Pareto-efficient".

---------------
Note: I identified several typos (not a weakness but a nontrivial issue to fix):
 * (Less serious) Line#292-#293:  "the reward of UniARM is computed as::" --> "the reward of UniARM is computed as:"
 * (Less serious) Line#429: "GenARM (Xu et al., 2025));" --> "GenARM (Xu et al., 2025);"
 * (More serious) Equation (5): There is a missing right parenthesis for the difference of the two log probabilities.
 * (More serious) also Equation (5): If I am not mistaken, either $(-1)^{z_i}$ needs to be negated as $-(-1)^{z_i}$ or the meaning of $z_i$ needs be flipped.

**Questions:**

see **Weakness** where I have included the questions.

---

> ### Author Response · Authors · 2025-11-19
>
> We sincerely thank Reviewer SHwo for the detailed and insightful feedback. Below, we provide detailed responses to the reviewer’s comments.
>
> > **Weaknesses 1:** Although the proposed method is empirically effective, there is a lack of intuitive or theoretical explanation of where the effectiveness (i.e., the better Pareto front) comes from. Can the authors explicitly provide such explanations? For example, why can a different parameterization of the token-level reward models alone (according to the ablation experiment when  $\lambda = 0$) lead to a better Pareto front?
> >
>
> We sincerely thank the reviewer for their constructive feedback. We would like to kindly refer you to Concern 1 of our global response, where we clarify the distinction between our method and prior methods, and provide an explanation of why, in the ablation experiment with $\lambda = 0$, a different parameterization of the token-level reward models can lead to a superior Pareto frontier.
>
> > **Weaknesses 2:** (Partially related to the first weakness) The generality of the method is unknown. Is the better Pareto front only limited to the Alpaca family? Can the authors provide results on other families (Llama, Qwen, etc.) to make the generality of the method more convincing?
> >
>
> We sincerely thank the reviewer for the kind reminder. We have supplemented experiments on other latest model families (Qwen-3 and Tulu-3) across different tasks and **would kindly like to refer you to Concern 2 of our global response for further details**.
>
> > **Weaknesses 3:** I find the term "Pareto-optimal" too strong and vague to describe an ARM as there is still room for improvement. I suggest a softer term "more/less Pareto-efficient".
> >
>
> Thank you for the suggestion. We have softened several expressions in the revised manuscript and replaced “Pareto-optimal” with “more Pareto-efficient” where appropriate.
>
> **typos**
>
> > (Less serious) Line#292-#293: "the reward of UniARM is computed as::" --> "the reward of UniARM is computed as:" (Less serious) Line#429: "GenARM (Xu et al., 2025));" --> "GenARM (Xu et al., 2025);"  (More serious) Equation (5): There is a missing right parenthesis for the difference of the two log probabilities.
> >
>
>
> Thank you for carefully reviewing our paper. We have already corrected the above typos in the updated manuscript.
>
> > (More serious) also Equation (5): If I am not mistaken, either  needs to be negated as  or the meaning of  needs be flipped.
> >
>
> We have already corrected the above typos in the updated manuscript. In fact, for the i-th preference dimension, we assign $z_i = 0$ if $y_1$ is preferred over $y_2$, and 1 otherwise. We sincerely apologize for those typos and greatly appreciate your careful reading of our paper.

---

> > ### Comment · Reviewer_SHwo · 2025-11-28
> >
> > Thank you for the rebuttal. As it addresses all my concerns, I will retain my favorable score.

---

> > > ### Author Response · Authors · 2025-11-28
> > >
> > > Thank you for your response! We are glad that your concerns have been addressed.

---

### Author Response · Authors · 2025-11-19
**Global Response**

We thank all reviewers for their valuable feedback.  The reviewers recognized our work for its strengths across multiple aspects:

- **Method Strengths**:
    - Recognized as **more reasonable** (rgLD)
    - Achieves **superior performance** (SHwo, 9Eti, rgLD, 97Hm)
    - Exhibits **higher parameter efficiency** (SHwo)
- **Paper Quality**:
    - **Well-written** (SHwo, rgLD)
    - **Experiments are well-designed / extensive** (SHwo, rgLD)
    - **Comprehensive related work** (9Eti, 97Hm)
    - **Intuitive case study presentation** (97Hm)
- **Implementation & Impact**:
    - **Easy to implement** (SHwo)
    - **Strong potential to significantly impact the community** (SHwo)
    - **Efficient alternative to training large LLMs** (rgLD)
    - Addresses an **interesting research direction** (97Hm)

Below, we address common concerns and misunderstandings and provide additional experimental results.

**Concern 1:**  The reviewers notes that MoSLoRA appears to be a minor modification of PBLoRA, as used in PARM, and requests an intuitive explanation for why MoSLoRA yields a superior Pareto frontier.

**Regarding differences between PBLoRA and MoSLoRA**  We respectfully clarify that MoSLoRA introduces a fundamental structural paradigm shift compared to PBLoRA [1], instead of a minor variant. **In PBLoRA, all modules are responsible for feature extraction, whereas in MoSLoRA, one module focuses on extracting shared features, and the other focuses on modulating these features based on preferences.**

- PBLoRA constructs k preference-specific LoRA branches on top of a preference-agnostic module $B_1 W_1 A_1$, where each branch  $B_2 W_{2,i} A_2$ learns features tailored to a specific preference:

$h_{\text{PBLoRA}} = (W_{\text{base}} + B_1 W_1 A_1 +   \sum_{i=1}^{k}  \alpha_i   B_2 W_{2,i} A_2 )h$

Because multiple branches operate on the same shared features and are ultimately aggregated at the output, no branch can independently specialize in its own preference-specific features, which in turn leads to feature entanglement across preferences.

- MoSLoRA **removes all preference-specific branches and introduce preference-modulation module ( $B_2 W_\gamma A_2$, $B_2 W_\eta A_2$)**. The preference-agnostic module and the preference modulation module have distinct roles and operate in separate spaces:

$h_{\text{MoSLoRA}} = (\gamma_{o'} + 1) \odot (W_{\text{base}} + B_1 W_1 A_1)h + \eta_{o'}$

$\gamma_{o'} = B_2 W_\gamma A_2 o'$

$\eta_{o'} = B_2 W_\eta A_2 o'$

The preference-agnostic module ($W_{\text{base}} + B_1 W_1 A_1$) learns shared features. The preference-modulation module  generates affine parameters ($\gamma_{o'}$, $\eta_{o'}$) conditioned on the mixed preference vector o’ to adjust the mean and scale of the shared features.

The preference-agnostic module is responsible for extracting shared features, while the modulation module does not learn features itself and only adjusts them at the output stage based on user preferences. This structural decoupling allows the learning of shared representations and the application of preference modulation to proceed independently, fundamentally preventing conflicts between the two modules.

**Regarding why MoSLoRA yields a better Pareto Frontier？**

PBLoRA relies on independent branches, which often produce discrete and fragmented solution regions. Features corresponding to different preferences frequently become entangled, resulting in a non-smooth Pareto frontier and reduced overall Pareto efficiency.

In contrast, MoSLoRA captures differences across preference combinations by applying affine transformations to the statistics (mean and scale) of the shared features, thereby constructing a continuous and adjustable objective manifold.

While PBLoRA learns feature distributions for each individual preference and then combines them, MoSLoRA directly learns the feature distribution corresponding to a specific combination of preferences. As a result, solutions under different preferences no longer rely on separate parameter subsets but are realized as smooth transformations of the shared feature space. This structural re-parameterization reduces feature interference and naturally produces a smoother and superior Pareto frontier. Furthermore, our method is simple and effective.

---

> ### Author Response · Authors · 2025-11-19
>
> **Concern 2: Verification of the approach on More Advanced Models and Across Different Model Families**
>
> We sincerely thank the reviewers for their kind and constructive feedback. We’ve added more model families,  **such as Qwen-3 [2] and Tulu-3** [3].
>
> - **We conducted experiments on the Safety Alignment task to evaluate the performance of our approach on the latest Qwen-3 family, following the same hyperparameter settings used for the Alpaca [4] family.** Specifically, Qwen3-8B-Base was fine-tuned to construct autoregressive reward models (GenARM [5], PARM [1], UniARM), which were then used to guide the generation of the frozen Qwen3-14B-Base model. The corresponding experimental results are presented below.
>
> Table1: Performance of GenARM, PARM , and UniARM on the safety alignment task. All methods are first fine-tuned on Qwen3-8B-Base, then guide the frozen Qwen3-14B-Base's generation.
>
> | Method | HV | MIP |
> | --- | --- | --- |
> | GenARM  | 182.43 | 5.23 |
> | PARM | 198.67 | 6.75 |
> | **UniARM** | **226.04** | **8.16** |
>
> As shown in Table 1, UniARM consistently outperforms PARM and GenARM on weak-to-strong generation tasks within the Qwen-3 model family, consistent with the trends reported for the Alpaca model family in the submitted manuscript.
>
> - **We conducted experiments on the helpful assistant task to evaluate the performance of our approach on the latest Tulu-3 family, following the same hyperparameter settings used for the Tulu-2 [6] family.** We fine-tuned GenARM, PARM, and UniARM on Tulu-3-8B and then used these models to guide the frozen Tulu-3-70B for generation. The results are presented below.
>
> Table2: Performance of GenARM, PARM , and UniARM on the  helpful assistant task. All methods are first fine-tuned on Tulu-3-8B, then guide the frozen Tulu-3-70B's generation.
>
> | Method | HV | MIP |
> | --- | --- | --- |
> | GenARM | 118.43 | 1.13 |
> | PARM | 133.51 | 1.46 |
> | **UniARM** | **147.29** | **1.73** |
>
> As shown in Table 2, when re-evaluated on the helpful assistant task using the Tulu-3 model family, all methods exhibit improved performance on weak-to-strong generation tasks; however, UniARM still outperforms GenARM and PARM, consistent with the observations in the Tulu-2 model family in the submitted manuscript.
>
> Across different model family (Alpaca, Tulu-2, Tulu-3, Qwen-3) and tasks (helpfulness, safety), UniARM consistently yields a strictly more Pareto-efficient, demonstrating strong generality. But we would also like to kindly remind the reviewers that in the *helpful assistant* task, we used **Tulu-2** and **Tulu-3**, which are fine-tuned versions of **Llama-2 [7]** and **Llama-3.1 [8]**, respectively. Therefore, the improved Pareto frontier also applies to the **Llama family**. **This indicates that the performance improvement of our approach does not rely on a specific backbone architecture**.
>
> [1] Lin B, Jiang W, Xu Y, et al. PARM: Multi-Objective Test-Time Alignment via Preference-Aware Autoregressive Reward Model[J]. ICML, 2025.
>
> [2] Yang A, Li A, Yang B, et al. Qwen3 technical report[J]. arXiv preprint arXiv:2505.09388, 2025.
>
> [3] Lambert N, Morrison J, Pyatkin V, et al. Tulu 3: Pushing frontiers in open language model post-training[J]. arXiv preprint arXiv:2411.15124, 2024.
>
> [4] Dai J, Pan X, Sun R, et al. Safe RLHF: Safe Reinforcement Learning from Human Feedback[C]//The Twelfth International Conference on Learning Representations.
>
> [5] Xu Y, Sehwag U M, Koppel A, et al. GenARM: Reward Guided Generation with Autoregressive Reward Model for Test-Time Alignment[C]. ICLR, 2025.
>
> [6] Ivison H, Wang Y, Pyatkin V, et al. Camels in a Changing Climate: Enhancing LM Adaptation with Tulu 2[J]. CoRR, 2023.
>
> [7] Touvron H, Martin L, Stone K, et al. Llama 2: Open foundation and fine-tuned chat models[J]. arXiv preprint arXiv:2307.09288, 2023.
>
> [8] Dubey A, Jauhri A, Pandey A, et al. The Llama 3 Herd of Models[J]. CoRR, 2024.

---

> ### Author Response · Authors · 2025-11-30
>
> **concern 3:**   Comparing and discussing different methods using ArmoRM-Llama3-8B-v0.1 **[1]** as the oracle reward model
>
> We sincerely thank the reviewer for the valuable comments and constructive feedback.
>
> To address this concern, we conducted additional experiments on the HelpSteer [2] dataset to evaluate how different methods perform when using ArmoRM-Llama3-8B-v0.1 as the oracle reward model.  HelpSteer focuses on promoting helpfulness, where we specifically examine two objectives: **correctness** (factual precision and relevance) and **verbosity** (response length and level of detail). This setup allows us to assess each method’s ability to navigate the trade-off between correctness and verbosity. We followed the same hyperparameter settings used for the safety alignment task.
>
> Table 3: Performance of RS [3], MOD [4], GenARM [5], PARM [6], and UniARM on the HelpSteer dataset. We fine-tuned GenARM, PARM, and UniARM on Qwen3-8B-Base, and then used these models to guide the generation of the frozen Qwen3-8B-Base. RS performs independent RL fine-tuning for each objective under the oracle reward, and then combines the resulting policies by interpolating their parameters in weight space. In contrast, MOD also performs independent RL fine-tuning for each objective, but generates the next token by taking a linear combination of the output predictions from all policy models.
>
> | **Method** | **HV** | **MIP** |
> | --- | --- | --- |
> | RS | 0.39 | 2.14 |
> | MOD | 0.45 | 2.33 |
> | GenARM | 0.57 | 2.35 |
> | PARM | 0.65 | 2.72 |
> | **UniARM** | **0.73** | **2.89** |
>
> As shown in Table 3, UniARM outperforms all baseline methods even when ARMO-RM is used as the oracle reward model. This result is fully consistent with the trends we observed in both the safety alignment and helpful assistant tasks. Specifically, in the safety alignment task, we employed beaver-7b-v1.0-reward and beaver-7b-v1.0-cost [7] as the oracle reward models; in the helpfulness assistant task, we used gpt2-large-helpful-reward_model, gpt2-large-harmless-reward_model [8], and [humor-no-humor](https://huggingface.co/mohameddhiab/humor-no-humor) as the oracle reward models.
>
> These additional experiments further confirm that UniARM demonstrates robust and consistent advantages across different oracle reward model settings.
>
> [1] Wang H, Xiong W, Xie T, et al. Interpretable Preferences via Multi-Objective Reward Modeling and Mixture-of-Experts[C]. ACL 2024
>
> [2] Wang Z, Dong Y, Zeng J, et al. Helpsteer: Multi-attribute helpfulness dataset for steerlm. ACL 2024
>
> [3] Rame A, Couairon G, Dancette C, et al. Rewarded soups: towards pareto-optimal alignment by interpolating weights fine-tuned on diverse rewards[J]. NIPS, 2023.
>
> [4] Shi R, Chen Y, Hu Y, et al. Decoding-time language model alignment with multiple objectives[J]. NIPS, 2024.
>
> [5] Xu Y, Sehwag U M, Koppel A, et al. GenARM: Reward Guided Generation with Autoregressive Reward Model for Test-Time Alignment[C]. ICLR, 2025.
>
> [6] Lin B, Jiang W, Xu Y, et al. PARM: Multi-Objective Test-Time Alignment via Preference-Aware Autoregressive Reward Model[J]. ICML, 2025.
>
> [7] Dai J, Pan X, Sun R, et al. Safe RLHF: Safe Reinforcement Learning from Human Feedback[C]//The Twelfth International Conference on Learning Representations.
>
> [8] Yang R, Pan X, Luo F, et al. Rewards-in-Context: Multi-objective Alignment of Foundation Models with Dynamic Preference Adjustment[C]. ICML, 2024.

---

### Public Comment · ~Taiqiang_Wu1 · 2025-11-23
**Missing Reference**

Dear Authors,

We are writing to introduce one related Paper#1: Mixture-of-Subspaces in Low-Rank Adaptation (https://arxiv.org/abs/2406.11909). This paper was accepted as EMNLP **2024** Oral. In this paper,  we proposed a method named as MoSLoRA. The method can be found at https://pic4.zhimg.com/v2-9ee7c1e437ce67822ad3042bad3b56ef_r.jpg  As you can see, the design is quite similar.

We acknowledge that this paper focus on a different task, i.e., multi-objective test-time alignment. Also, searching all the related work is quite hard nowadays. However, we think a discussion is required.

Best,

Paper #1 Authors

---

> ### Author Response · Authors · 2025-11-24
>
> We appreciate your kind reminder and recognize the superficial similarity in the name. We were not aware of your work before and apologize for missing it in our earlier manuscript. We would like to clarify that our main design focuses on modulating shared features within a unified parameter space to accommodate different preference combinations. Therefore, although the two methods share the same abbreviation, the structure and functionality of our MoSLoRA (UniARM) are fundamentally different from yours. We will make sure to cite and discuss your work in the updated manuscript.

---

### Meta-Review · Area_Chair_KFpn · 2026-01-05

**Summary:**

The authors’ response to the AC is professional and responsive—scores moved in the right direction (SHwo/rgLD) and the added ArmoRM evaluation is a reasonable attempt to close the loop for 9Eti/97Hm—but the core interpretability gap that drives the split reviews remains: the paper talks like it has a principled handle on Pareto efficiency over preference representations, yet never states (let alone defends) a credible hypothesis for what “optimal” representation even means at the level of a single input–response pair, much less how one would certify coverage against a true oracle frontier across sub-populations. SHwo’s request for an intuitive/theoretical explanation of why the parameterization alone yields a better front, and 97Hm’s concern that gains may simply reflect reliance on a predefined semantic representation or missing comparisons that anchor the new problem to standard alignment, are not discharged by the rebuttal’s architectural narrative (“shared features + affine modulation”), which reads more as a post-hoc mechanism sketch than an argument with falsifiable implications. In short: the rebuttal does real work on breadth and benchmarking, and the AC reply appropriately emphasizes that effort, but the manuscript still over-claims “strictly more Pareto-efficient” preference modeling without communicating the requisite coverage/optimality conditions under which such a claim would be meaningful or verifiable.

**Reviewer Concerns:**

Reviewer concerns resolved by the rebuttal

Limited evaluation on outdated model families
Reviewers (SHwo, 9Eti, rgLD, 97Hm) questioned whether results generalize beyond Alpaca-era models; the rebuttal adds Qwen-3 and Tulu-3 experiments, which multiple reviewers (SHwo, rgLD) explicitly acknowledge as resolving this concern.

Lack of comparison using stronger / modern oracle reward models
Reviewers (9Eti, 97Hm) requested evaluation with ArmoRM; the rebuttal adds ArmoRM-Llama3-8B-v0.1 experiments showing consistent gains, directly addressing this request.

Insufficient experimental breadth and ablations
Reviewers (SHwo, rgLD) asked for stronger empirical support; the rebuttal expands tasks, model scales, weak-to-strong settings, and hyperparameter analyses, which SHwo confirms as fully addressed.

Overly strong wording around “Pareto-optimality”
Reviewer SHwo flagged terminology as too strong; the authors revise the manuscript to use “more Pareto-efficient,” resolving the issue.

Missing implementation details and minor correctness issues
Reviewers (SHwo, 9Eti, rgLD, 97Hm) noted typos, undefined notation, and missing cost/runtime details; these were corrected or clarified in the revised manuscript.

**Reviewer Scores:**

See above.

---

### Decision · Program_Chairs · 2026-01-26

Reject